# Synthesis of 6-Halo-Substituted Pericosine A and an Evaluation of Their Antitumor and Antiglycosidase Activities

**DOI:** 10.3390/md20070438

**Published:** 2022-06-30

**Authors:** Yoshihide Usami, Yoshino Mizobuchi, Mai Ijuin, Takeshi Yamada, Mizuki Morita, Koji Mizuki, Hiroki Yoneyama, Shinya Harusawa

**Affiliations:** 1Department of Pharmaceutical Organic Chemistry, Osaka University of Pharmaceutical Sciences, Nasahara 4-20-1, Takatsuki 569-1094, Osaka, Japan; e16101@gap.oups.ac.jp (Y.M.); e14509@gly.oups.ac.jp (M.I.); e12546@gap.oups.ac.jp (M.M.); e12007@gap.oups.ac.jp (K.M.); hiroki.yoneyama@ompu.ac.jp (H.Y.); harusawa@gly.oups.ac.jp (S.H.); 2Department of Medicinal Molecular Chemistry, Osaka University of Pharmaceutical Sciences, Nasahara 4-20-1, Takatsuki 569-1094, Osaka, Japan; takeshi.yamada@ompu.ac.jp

**Keywords:** synthesis, 6-halogenated pericosine A analogues, antitumor, glycosidase inhibition, enantiomer, pericoxide

## Abstract

The enantiomers of 6-fluoro-, 6-bromo-, and 6-iodopericosine A were synthesized. An efficient synthesis of both enantiomers of pericoxide via 6-bromopericosine A was also developed. These 6-halo-substituted pericosine A derivatives were evaluated in terms of their antitumor activity against three types of tumor cells (p388, L1210, and HL-60) and glycosidase inhibitory activity. The bromo- and iodo-congeners exhibited moderate antitumor activity similar to pericosine A against the three types of tumor cell lines studied. The fluorinated compound was less active than the others, including pericosine A. In the antitumor assay, no significant difference in potency between the enantiomers was observed for any of the halogenated compounds. Meanwhile, the (−)-6-fluoro- and (−)-6-bromo-congeners inhibited α-glucosidase to a greater extent than those of their corresponding (+)-enantiomers, whereas (+)-iodopericosine A showed increased activity when compared to its (−)-enantiomer.

## 1. Introduction

Chemical modification of bioactive natural products is one of the most common methods used in drug development [1,2,3,4,5,6,7,8]. In addition to increasing the pharmacological activity, which is the major aim of chemical modification, decreasing side effects, improving the solubility or stability, and reducing costs remain problems to be resolved during drug development. Chemical modification based on marine natural products has been extensively studied due to the discovery of new lead compounds suitable for drug discovery [9,10,11,12]. 

The isolation of pericosine A (**1_Cl_**) and B (**2**) as metabolites of the marine-derived fungus *Periconis byssoides* N133 was first reported in 1997, and pericosines C–E (**3**–**6**) were discovered in 2008 [13,14,15]. The unique carbasugar structures constituting the highly functionalized cyclohexene ring are shown in Figure 1. Recently, the chemistry of carbasugars has received an increasing amount of research attention due to their wide range of biological activity [16,17,18]. In addition, many total syntheses of pericosines A–C (**1**–**3**) have been reported due to the antitumor activity of pericosine A (**1_Cl_**) [19,20,21,22,23,24,25,26,27,28,29,30]. However, the chemical modification of the pericosines has not been reported to date with the exception of our synthetic study on pericosine E analogs bearing chloro- or methoxy-substituents at C-6, which were used as α-glucosidase inhibitors [31,32]. Figure 1 also includes (+)-pericoxide (**7**) and (−)-maximiscin, which were discovered by Cichewitcz and coworkers from *Tolypocladium* sp. [33,34]. The fact that pericosine **1_Cl_** was obtained together with pericoxide **7** in their work suggests that **7** may be a biosynthetic precursor of **1****_Cl_**. Therefore, **7** should be classified as a member of the pericosine family.

In our recent report on pericosine A, natural **1_Cl_** was isolated from *Periconia* sp. as an enantiomeric mixture. In addition, we confirmed that both synthesized enantiomers of **1_Cl_** exhibit antitumor activity with similar potency against three tumor cell lines (P388, LH-6, and L1210). Furthermore, (−)-**1_Cl_** showed moderate α-glucosidase inhibitory activity (IC_50_ = 2.25 mM), whereas (+)-**1_Cl_** was inactive [35]. As we are interested in the biological activities of pericosine congeners bearing other halogen atoms, the fermentation of *Periconia* sp in artificial seawater containing fluoride, bromide, or iodide sources has been examined as an alternative to chloride. It should be mentioned here that 6-halogenated pericosines bearing F, Br, or I atoms were not obtained in our preceding paper. Consequently, we have attempted the synthesis of non-natural 6-halo-substituted pericosine A. Our recent synthetic work on pericosine E analogs has suggested that the presence of a chlorine atom at C6 is an important factor for α-glucosidase inhibitory activity. The preparation of other 6-halo-congeners is also required for our continuous synthetic studies on new pericosine E analogs. 

Herein, we describe the synthesis and evaluation of the antitumor and antiglycosidase inhibitory activities of both enantiomers of newly designed 6-halo-congeners of pericosine A. In addition, an efficient synthesis of closely related pericoxide **7** via 6-bromo pericosine A (**1**_Br_) has also been reported. 

## 2. Results & Discussion

### 2.1. Synthesis of Both Enantiomers of 6-Halo-Substituted Pericosine A

A similar reaction to the hydrochlorination of epoxide intermediate **8** used in the synthesis of pericosine A **1_Cl_** [21,29] using commercially available HBr and HI aqueous solutions was envisioned in this study, in which the deprotection of the hydroxyl groups and hydrolysis of the ester moiety will occur (Figure 1). Subsequently, we began this work by searching for suitable reagents used to introduce the required fluorine, bromine, or iodine atoms.

For bromination, BBr_3_ was initially examined as the bromide source to react with epoxide (−)-**8**, which can be prepared from bromohydrin (+)-**9** via an intramolecular S_N_2 reaction [29,31]. However, careful addition of BBr_3_ [0.33 equivalents (eq)] to (−)-**8** in dry diethyl ether (Et_2_O) at −78 °C afforded the desired product (**10**_Br_) in only 14% yield. Using 1.0 eq of BBr_3_ slightly improved the yield (38%), but increasing the reaction temperature led to the formation of a complicated mixture including the undesired regioisomer. After investigating a variety of brominating reagents*,* we found mono-bromoborane dimethyl sulfide complex (BH_2_Br·SMe_2_) to be the most suitable for the desired reaction [36,37]. The reaction of **8** with 1.0 eq of BH_2_Br·SMe_2_ at −78 °C in Et_2_O dramatically improved the yield of (−)-**10**_Br_ to 94% yield. The reactions performed at higher temperatures led to a decreased yield of (−)-**10**_Br_ (78% at −20 °C and 58% at 0 °C). Subsequent deprotection of the cyclohexylidene moiety was also a delicate process. The optimum reaction conditions were found after investigating a variety of conditions. Treatment of (−)-**10**_Br_ with Dowex^®^ 50WX8 hydrogen form (Dowex^®^ 50WX8-H: Acidic ion-exchange resin) in MeOH at room temperature (rt) for 56 h gave (−)-**1**_Br_ in an excellent 87% yield when compared to the conventional reaction using trifluoroacetic acid (TFA) in MeOH (66%) (Figure 1A). This deprotection process with Dowex-XW50-H was examined in detail because could also be applied in the subsequent synthesis of pericoxide **7** (see Appendix A).

The corresponding (+)-enantiomer of **1**_Br_ was prepared in the same way using (+)-**8** derived from (−)-quinic acid (Figure 1B). Iodohydrination of (−)-**8** was then achieved upon careful addition of 0.33 eq of aluminum iodide (AlI_3_) at 0 °C to give the desired product [(−)-**10**_I_] in 63% yield [38]. When 1.0 eq of AlI_3_ was used under the same reaction conditions, the yield of (−)-**10**_I_ was reduced to 15%. The reaction of (−)-**8** with 1.0 eq of tetrabutylammonium iodide and a catalytic amount of BF_3_·Et_2_O in dry dichloromethane at −78 °C afforded (−)-**10**_I_ in 17% yield. Finally, deprotection of (−)-**10_I_** was achieved under conventional TFA/MeOH conditions to give (−)-**1**_I_ in 76% yield, whereas treatment of (−)-**10_I_** with Dowex^®^ 50WX8-H resulted in no reaction. The (+)-enantiomer of **1**_I_ was prepared using a similar approach from (+)-**8** via (+)-**10**_I_.

The introduction of a fluorine atom into (−)-**8** was accomplished using the (HF)_n_/py complex. The reaction of (−)-**8** with this complex at 0 °C for 15 min in a polypropylene tube afforded the desired fluorohydrine product [(−)-**10**_F_] in 46% yield. A longer reaction time (1 h) led to a more complex product mixture, giving a low yield of (−)-**10**_F_ (32%). Subsequent treatment of (−)-**10**_F_ with TFA in MeOH afforded (−)-**1**_F_ in 43% yield. Similarly, (+)-**1**_F_ was successfully obtained using (+)-**8**. It should be noted that **10**_I_ and **1**_I_ are relatively unstable when compared to the other halogenated congeners, so they were stored in a freezer prior to further use.

### 2.2. Synthesis of Pericoxide

The synthesis of **7**, which was not reported prior to 2019, was also conducted, as shown in Figure 2. The direct deprotection of epoxide **8** was examined in our preliminary efforts to prepare **7**. Treatment of (−)-**8** with trifluoroacetic acid in *t*-BuOH at room temperature (rt) did not afford pericoxide **7** despite the complete consumption of **8**. Changing the acid catalyst to Dowex^®^ 50WX8-H resulted in no reaction and microwave (MW) heating afforded a small amount of methyl 3,4-dihydroxybenzoate along with the recovery of the starting material [(−)-**8**].

In our next experiment, bromotriol **11** derived from bromohydrin **9** was treated with 3.0 eq of lithium hexamethyldisilazide (LHMDS) in THF at −78 °C because if the C6 hydroxide anion attacked the C5 center faster than the C4 hydroxide anion, the formation of **7** via an intramolecular S_N_2 process was expected to occur. However, the spectral data of epoxide **12** did not agree with those of **7**. The structure of **12** was determined using NMR spectroscopy; epoxy carbon atoms C6 (δ 55.5 ppm) and C1 (δ 56.0 ppm) were detected in the high field region in the ^13^C-NMR spectrum and the HMBC cross peaks corresponding to H1/C3 and H6/C4 indicated the structure of **12** (see Appendix A).

Finally, (−)-bromopericosine A (**1_Br_**) was treated with 3.0 eq of LHMDS at −78 °C to give (−)-**7**, which is an enantiomer of the natural product. The isolated yield of (−)-**7** upon purification via conventional acidic silica gel column chromatography was only 23%, but this was improved to 77% using neutral silica gel. The spectral data agree with those of natural pericoxide with the exception of the sign of its specific rotation. Unfortunately, (−)-**7** was so unstable that it decomposed upon storage at rt in methanol, exhibiting a smaller optical rotation within a couple of days. Furthermore, the decomposed residue exhibits a positive specific rotation, although freshly synthesized **7** was negative. Regrettably, biological assays of **7** could not be performed because of this inherent instability.

Treatment of **1**_Br_ with 2.0 eq of LHMDS led to a decreased yield of **7** (57%) along with the recovered starting material (**1**_Br_; 34%). Alternatively, intramolecular epoxidation of (−)-**1_Cl_** with LHMDS (3.0 eq) gave (−)-**7** in 12% yield along with the recovery of (−)-**1_Cl_** (20%).

This experiment indicates the significance of bromo-derivative **1**_Br_ as the starting material. Using the same synthetic process, natural (+)-**7** was synthesized via (+)-**1_Br_** starting from (−)-quinic acid**_._** A similar synthesis of (+)-**7** from (+)-**1_Cl_** has been previously reported by Cichewitz in 2019 [39].

### 2.3. Evaluation of the Biological Activities of Enantiomerically Pure Pericosine A and Its 6-Halogenated Congeners

#### 2.3.1. Antitumor Assay

The antitumor activity of halo-compounds **1** was evaluated against three types of tumor cell lines: Basic P388 (mouse lymphocytic leukemia), L1210 (mouse lymphocytic leukemia), and LH60 (human promyelocytic leukemia) cell lines, along with a previously reported procedure using 5-fluorouracil (5-FU) as a positive control [35]. The results are presented in Table 1, including those obtained for perocosine A (**1_Cl_**), which has been previously reported in the literature, for comparison. All compounds showed antitumor activity against the three types of tumor cell lines studied. Bromo- and iodo-pericosine (**1_Br_** and **1**_I_) show similar activities to **1_Cl_**, but fluorinated compound **1**_F_ was less active than all of the other compounds, including **1_Cl_**. Because pericosine C (**3**), which exists as an enantiomeric mixture in nature and has the same relative configuration to **1**s, was reported to be inactive against P-388 cell line [14], present results implied the importance of the presence of halogen atom at C-6 in pericosine core structure for antitumor activity. In addition, it is noteworthy that no significant difference in the potency was observed between the enantiomers, similar to pericosine A. 

#### 2.3.2. Glycosidase Inhibitory Activity Assay 

There have been several excellent studies on the development of pseudosugar-type glycosidase inhibitors [40,41,42,43,44,45,46,47,48,49,50]. The compounds synthesized in this study were also applied to an glycosidase inhibitory assay against five kinds of enzymes: α-Glucosidase from yeast, β-glucosidase from Jack bean, α-galactosidase from green coffee bean, β-galactosidase from bovine liver, and α-mannosidase from bovine liver. The results are presented in Table 2 along with the previous results obtained for pericosine A (**1_Cl_**) for comparison [35]. (−)-**1**_F_, (−)-**1**_Br_, and (−)-**1**_I_ exhibit comparable inhibitory activities to (−)-**1_Cl_** (IC_50_ = 2.25 mM) against α-glucosidase with IC_50_ values of 1.95, 1.79, and 3.60 mM, respectively, whereas they were less active than the positive control [deoxynojirimycin (DNJ); IC_50_ = 0.0965 mM]. (+)-**1_Br_** showed less potent activity (IC_50_ = 5.05 mM) when compared to (−)-**1**_Br_, and (+)-**1**_F_ was inactive similar to (+)-**1**_Cl_. Surprisingly, (+)-**1_I_** exhibited the most potent α-glucosidase inhibitory activity (IC_50_ = 1.15 mM) among the eight compounds studied and was active against α-galactosidase (IC_50_ = 3.56 mM). This is the only example of an α−galactosidase inhibitor among the pericosines and their congeners reported to date.

All of the synthesized compounds were inactive against β-glucosidase and α-galactosidase. Interestingly, (−)-**1**_Br_ showed dual activity against β-galactosidase and α-glucosidase similar to (−)-**1_Cl_**, but they were less active than those reported for some pericosine E derivatives [33].

## 3. Materials and Methods

General methods: HRMS was performed on a JMS-700 (2) mass spectrometer (JEOL, Tokyo, Japan). NMR spectra were recorded at 27 °C on 300- and 400-MR-DD2, INOVA-500 and 600-DD2 spectrometers (Agilent Technologies, CA, USA) in CDCl_3_ or acetone-d_6_ using tetramethylsilane (TMS) as an internal standard. Specific rotations were measured using a DIP1000 digital polarimeter (JASCO Co., Tokyo, Japan). Liquid column chromatography was conducted on silica gel (BW-127ZH) (Fuji Silysia, Tokyo, Japan) or neutral silica gel (CHROMATOREX DIOL MB100–75/200) (Fuji Silysia, Tokyo, Japan). Analytical TLC was performed on precoated silica gel 60 plates (Merck & Co., Inc., Darmstadt, Germany) and the compounds were viewed by dipping the plates in an ethanol solution of phosphomolybdic acid, followed by heating. Microwave-aided reactions were performed using Initiator^®^ (Biotage, Uppsala, Sweden). Flash chromatography was performed using Isolera One^®^ (Biotage, Uppsala, Sweden). (−)-Shikimic acid was purchased from Carbosynth Ltd. (UK). (−)-Quinic acid was purchased from Merck & Co., Inc. (Darmstadt, Germany). AlI_3_, HCl, BH_2_Br, SMe_2_, and Dowex^®^ 50WX8-H were purchased from Sigma-Aldrich (St. Louis, MO). n-BuLi in hexane was purchased from Nacalai Tesque (Kyoto, Japan). Hexamethyldisilazane and (HF)_n_/py complex were purchased from TCI (Tokyo, Japan). Trifluoroacetic acid, dry MeOH, CH_2_Cl_2_, Et_2_O, and tetrahydrofuran (THF) were purchased from Wako Pure Chemical Industries (Osaka, Japan).

### 3.1. Synthesis of Both Enantiomers of The 6-Halopericosine A Analogs

#### 3.1.1. Synthesis of Methyl (−)-3,4-O-Cyclohexylidene-6-Fluoro-3,4,5-Trihydroxy-1-CycloHexene Carboxylate (10_F_) 

To a solution of (−)-**8** (13.4 mg, 0.050 mmol) in CH_2_Cl_2_ (1.0 mL) in a polyethylene tube and cooled at 0 °C was added 12.5 mL of (HF)_n_/pyridine complex (67% w/v HF, 4.0 mmol). The resulting mixture was stirred at 0 °C for 15 min and quenched upon adding saturated (sat.) NaHCO_3_ (aq) (10 mL). The mixture was then extracted with CH_2_Cl_2_ (3 × 10 mL). The combined organic layers were dried over MgSO_4_, filtered, and evaporated to afford the crude product, which was purified by column chromatography on neutral silica gel (eluent = Hexane:EtOAc = 9:1) to afford (−)-**10**_F_ as a colorless oil (6.6 mg, 46%). [α]_D_^20^ −41.1 (c 0.12, CHCl_3_); ^1^H-NMR (CDCl_3_, 500 MHz) δ 1.22–1.71 (10H, m), 2.51 (1H, dd, *J* = 4.1 1.6 Hz, 5-OH), 3.83 (3H, s, -COOCH_3_), 4.20 (1H, dddd, *J* = 10.3, 5.5, 4.5, 3.2 Hz, H-5), 4.56 (1H, ddd, *J* = 6.2, 3.2, 3.0 Hz, H-4), 4.76 (1H, dddd, *J* = 6.2, 5.5, 3.2, 0.9 Hz, H-3), 5.56 (1H, dd, *J* = 48.0, 5.5 Hz, H-6), 7.01 (1H, br dd, *J* = 3.2, 2.8 Hz, H-2); ^13^C-NMR (CDCl_3_, 125 MHz) δ 23.6 (CH_2_), 23.9 (CH_2_), 24.9 (CH_2_), 34.5 (CH_2_), 36.6 (CH_2_), 52.4 (CH_3_, CO_2_*C*H_3_), 67.6 (d, ^2^*J*_C-F_ = 24.9 Hz, CH, C-5), 70.9 (d, ^4^*J*_C-F_ = 2.4 Hz, CH, C-3), 73.4 (d, ^3^*J*_C-F_ = 5.8 Hz, CH, C-4), 86.2 (d, ^1^*J*_C-F_ = 168.8 Hz, CH, C-6), 111.2, 128.6 (d, ^2^*J*_C-F_ = 17.7 Hz, CH, C-1), 139.4 (d, ^3^*J*_C-F_ = 5.8 Hz, CH, C-2), 165.2 (Cq, *C*OOMe); HREIMS *m/z* calcd for C_14_H_19_O_5_F [M]^+^ 286.1217, found 286.1218.

(+)-**10_F_** (7.1 mg, 43%) was synthesized from (+)-**8** (15.5 mg, 0.058 mmol) in the same manner as (−)-**10_F_**. [α]_D_^20^ +37.6 (*c* 0.13, CHCl_3_); ^1^H-NMR (CDCl_3_, 400 MHz) δ 1.22–1.70 (10H, m), 2.51 (1H, d, *J* = 3.0 Hz, 5-OH), 3.83 (3H, s, -COOCH_3_), 4.17–4.24 (1H, m, H-5), 4.56 (1H, dt, *J* = 6.3, 3.1 Hz, H-4), 4.76 (1H, ddd, *J* = 6.0, 5.5, 3.1 Hz, H-3), 5.56 (1H, dd, *J* = 48.1, 5.5 Hz, H-6), 7.01 (1H, br dd, *J* = 2.8, 2.5 Hz, H-2); ^13^C-NMR (CDCl_3_, 100 MHz) δ 23.6, 23.9, 24.9, 34.5, 36.6, 52.4, 67.6 (d, ^2^*J*_C-F_ = 25.2 Hz, C-5), 70.9 (d, ^4^*J*_C-F_ = 2.3 Hz, C-3), 73.4 (d, ^3^*J*_C-F_ = 5.4 Hz, C-4), 86.2 (d, ^1^*J*_C-F_ = 168.6 Hz, C-6), 111.2 (Cq), 128.5 (d, ^2^*J*_C-F_ = 17.6 Hz, CH, C-1), 139.4 (d, ^3^*J*_C-F_ = 5.3 Hz, C-2), 165.2 (Cq, *C*OOMe); HREIMS *m/z* calcd for C_14_H_19_O_5_F [M]^+^ 286.1217, found 286.1215.

#### 3.1.2. Synthesis of (−)-6-Fluoropericosine A (1_F_) 

To a solution of (−)-**10**_F_ (7.5 mg, 0.026 mmol) in MeOH (1.0 mL) was added TFA (2.0 mL) at 0 °C with stirring. The reaction mixture was stirred for another 3.5 h at rt, then condensed under reduced pressure give the crude product, which was purified by column chromatography on neutral silica gel (eluent = 5% MeOH in CH_2_Cl_2_) to afford (−)-**1_F_** as a colorless oil. [α]_D_^20^ −78.7 (*c* 1.4, MeOH); ^1^H-NMR (MeOH-d_4_, *600 MHz*, ppm) δ 3.76 (3H, s, -OMe), 3.97 (1H, ddd, *J* = 16.7, 6.1, 2.0 Hz, H-5), 4.03–4.06 (1H, m, H-4), 4.41–4.44 (1H, m, H-3), 5.42 (1H, dddd, *J* = 49.6, 6.7, 1.7, 0.6 Hz, H-6), 6.81–6.82 (1H, m, H-2); ^13^C-NMR (MeOH-d_4_, *150 MHz*, ppm) δ 52.3 (CH_3_, COO*Me*), 68.2 (CH, C3, ^4^*J*_C-F_ = 2.3 Hz), 72.7 (CH, C4, ^3^*J*_C-F_ = 6.9 Hz), 73.3 (CH, C5, ^2^*J*_C-F_ = 20.9 Hz), 90.7 (CH, C-6, d, ^1^*J*_C-F_ = 167.6 Hz), 129.1 (Cq, C1, d, ^2^*J*_C-F_ = 18.5 Hz), 144.4 (CH, C-2, d, ^3^*J*_C-F_ = 5.8 Hz), 166.3 (Cq, *C*OOMe); HREIMS *m*/*z* calcd for C_8_H_12_O_5_F [M]^+^ 207.0669, found 207.0668.

(+)-**1_F_** (3.5 mg, 59%) was synthesized from (+)-**10_F_** (8.2 mg, 0.029 mmol) in the same manner as (−)-**1_F_**. Colorless oil; [α]_D_^20^ +67.7 (*c* 0.95, MeOH); H-NMR (MeOH-d_4_, *600 MHz*, ppm) δ 3.73 (3H, s, -OMe), 3.92 (1H, ddd, *J* = 17.0, 6.1, 2.0 Hz, H-5), 4.03–4.05 (1H, m, H-4), 4.40–4.43 (1H, m, H-3), 5.34 (1H, dddd, *J* = 49.9, 6.5, 2.1, 0.6 Hz, H-6), 6.75–6.77 (1H, m, H-2); ^13^C-NMR (MeOH-d_4_, *150 MHz*, ppm) δ 52.4 (CH_3_, COO*Me*), 68.3 (CH, C3), 73.0 (CH, C4, ^3^*J*_C-F_ = 6.9 Hz), 73.1 (CH, C5, ^2^*J*_C-F_ = 20.8 Hz), 90.8 (CH, C-6, d, ^1^*J*_C-F_ = 167.6 Hz), 129.0 (Cq, C1, d, ^2^*J*_C-F_ = 18.5 Hz), 144.4 (CH, C-2, d, ^3^*J*_C-F_ = 6.9 Hz), 166.5 (Cq, *C*OOMe); HREIMS *m*/*z* calcd for C_8_H_12_O_5_F [M]^+^ 207.0669, found 207.0675.

#### 3.1.3. Synthesis of Methyl (−)-6-Bromo-3,4-O-Cyclohexylidene-3,4,5-Trihydroxy-1-CycloHexene Carboxylate (10_Br_)

To a solution of *syn*-epoxide (−)-**8** (112.8 mg, 0.42 mmol) in Et_2_O (5 mL) was added a solution 1.0 M BH_2_Br·SMe_2_ in CH_2_Cl_2_ (0.48 mL, 0.42 mmol) at −78 °C. After stirring at −78 °C for 5 h, the reaction mixture was quenched with saturated NH_4_Cl aq., and extracted with CH_2_Cl_2_ (10 mL × 3). The combined organic layers were dried over MgSO_4_, filtered, and evaporated to give the crude product, which was almost pure, but purified via silica gel column chromatography (eluent; EtOAc:Hexane = 1:3) to afford (−)-**10**_Br_ (139.7 mg, 94%). [α]_D_^20^ −207.4 (*c* 0.30, CHCl_3_); IR (KBr) ν_max_ 3471 (OH), 1724 (C=O), 1652 (C=C) cm^−1^; ^1^H-NMR (acetone-*d_6_*, 600 MHz, ppm) δ 1.37–1.71 (10H, m, 5 × CH_2_), 3.80 (3H, s, -OMe), 4.17 (1H, d, *J* = 4.1 Hz, OH), 4.29 (1H, ddd, *J* = 4.4, 4.1, 3.5 Hz, H-5), 4.71 (1H, br dd, *J* = 7.0, 3.5 Hz, H-4), 4.88 (1H, br dd, *J* = 7.0, 3.2 Hz, H-3), 4.99 (1H, br d, *J* = 4.4 Hz, H-6), 6.99 (1H, dd, *J* = 3.2, 1.5 Hz, H-2); ^13^C-NMR (acetone-*d_6_*, *150 MHz*, ppm) δ 24.4 (CH_2_), 24.7 (CH_2_), 25.8 (CH_2_), 34.91 (CH_2_), 34.93(CH_2_), 36.8 (CH_2_), 44.0 (CH, C-6), 52.6 (Cq, -O*C*H_3_), 68.9 (CH. C-5), 70.8 (CH, C-3), 73.6 (CH, C-4), 111.1 (Cq), 131.6 (Cq, C-1), 138.6 (CH, C-2), 165.8 (Cq, *C*OOMe); ^1^H-NMR (CDCl_3_, *600 MHz*, ppm) δ 1.40–1.73 (10H, m, 5 × CH_2_), 2.70 (1H, d, *J* = 2.3 Hz, OH), 3.83 (3H, s, -OMe), 4.35–4.37 (1H, m, H-5), 4.75–4.79 (2H, m, overlapped, H-3,4), 4.88 (1H, br dd, *J* = 7.0, 3.2 Hz, H-3), 5.09 (1H, d, *J* = 3.8 Hz, H-6), 7.17 (1H, dd, *J* = 1.8, 1.1 Hz, H-2); ^13^C-NMR (CDCl_3_, *150 MHz*, ppm) δ 23.5 (CH_2_), 23.9 (CH_2_), 25.1 (CH_2_), 33.3 (CH_2_), 36.0 (CH_2_), 40.4 (CH-C6), 52.5 (Cq, -O*C*H_3_), 67.0 (CH, C-5), 69.4 (CH, C-3), 71.6 (CH, C-4), 110.7 (Cq), 130.7 (Cq, C-1), 137.4 (CH, C-2), 164.7 (Cq, *C*OOMe): C_14_H_19_O_5_^81^Br [M]^+^ 348.0395 found 348.0395. HREIMS *m/z* calcd for C_14_H_19_O_5_^79^Br [M]^+^ 346.0416 found 346.0415, C_14_H_19_O_5_^81^Br_1_ (M)^+^ 348.0395 found 348.0395.

(+)-**10**_Br_ (57.6 mg, 83%) was synthesized from (+)-**8** (53.2 mg, 0.20 mmol) in the same manner as (−)-**10_Br_**. Colorless crystals (CH_2_Cl_2_); mp 117–120 °C; [α]_D_^20^ +204.0 (*c* 0.29, CHCl_3_); ^1^H-NMR (acetone-d_6_, *600 MHz*, ppm) δ 1.38–1.72 (10H, m, 5 × CH_2_), 3.81 (3H, s, OMe), 4.19 (1H, d, *J* = 4.1 Hz, 5-OH), 4.27–4.30(1H, m, H-5), 4.71 (1H, br dd, *J* = 7.0, 3.5 Hz, H-4), 4.88 (1H, dd, *J* = 7.0, 2.9 Hz, H-3), 4.99 (1H, d, *J* = 4.4 Hz, H-6), 6.99 (1H, dd, *J* = 3.3, 1.5 Hz, H-2); ^13^C-NMR (acetone-d_6_, *150 MHz*, ppm) δ 24.4 (CH_2_), 24.7 (CH_2_), 25.8 (CH_2_), 34.9 (CH_2_), 36.8 (CH_2_), 44.0 (CH, C-6), 52.6 (CH_3_, COO*Me*), 68.9 (CH, C5), 70.8 (CH, C3), 73.6 (CH, C4), 111.1 (Cq), 131.6 (Cq, C1), 138.7 (CH, C-2), 165.8 (Cq, *C*OOMe); HREIMS *m*/*z* calcd for C_14_H_19_O_5_^79^Br [M]^+^ 346.0416, found 346.0415, C_14_H_19_O_5_^81^Br [M]^+^ 348.0395, found 348.0395.

#### 3.1.4. Synthesis of (−)-6-Bromopericosine A (1_Br_)

Appendix A, entry 4: To a solution of (−)-**10**_Br_ (41.3 mg, 0.12 mmol) in MeOH (2.0 mL) was added Dowex^®^ 50WX8-H (102.5 mg) and the resulting mixture was stirred for 56 h at room temperature. The reaction mixture was filtered and the filtrate concentrated under reduced pressure to give a crude residue, which was purified with silica gel column chromatography (eluent; CH_2_Cl_2_:MeOH = 95: 5) to afford (−)-**1**_Br_ (27.6 mg, 87%) and recovered (−)-**10**_Br_ (5.5 mg, 13%). (−)-**1**_Br_: Amorphous solid; mp 96–99 °C; [α]_D_^20^ −130.8 (*c* 1.38, MeOH); IR (film) ν_max_ 3219 (OH), 1712 (C=O) cm^−1^; ^1^H-NMR (CD_3_OD, *600 MHz*, ppm) δ 3.80 (3H, s, COOMe), 4.10 (1H, dd, *J* = 4.4, 2.1 Hz, H-4), 4.18 (1H, ddd, *J* = 4.4, 2.0 Hz, H-5), 4.36 (1H, br dd, *J* = 4.4, 4.1Hz, H-3), 4.94 (1H, dd, *J* = 4.4, 0.9 Hz, H-6), 6.86 (1H, d, *J* = 4.1 Hz, H-2); ^13^C-NMR (CD_3_OD, *150 MHz*, ppm) δ 48.7 (CH, C-6), 52.8 (CH_3_, COO*Me*), 67.6 (CH, C-3), 69.3 (CH, C-4), 76.2 (CH, C-5), 131.8 (Cq, C-1), 141.6 (CH, C-2), 167.3 (Cq, *C*OOMe); HREIMS *m/z* calcd for C_8_H_12_O_5_^79^Br [M + H]^+^ 266.9868 found 266.9869.

(+)-**1**_Br_ (37.7 mg, 78%) was synthesized from (+)-**10**_Br_ (57.6 mg, 0.166 mmol) in the same manner as (−)-**1**_Br_. Amorphous solid; mp 96–99 °C; [α]_D_^20^ +130.2 (*c* 1.74, MeOH); ^1^H-NMR (MeOH-d_4_, *600 MHz*, ppm) δ 3.80 (3H, s, -OMe), 4.10 (1H, *J* = 4.4, 2.1 Hz, H-5), 4.10 (1H, dd, *J* = 4.4, 2.1 Hz, H-4), 4.36 (1H, br dd, *J* = 4.7, 4.4 Hz, H-3), 4.94 (1H, dd, *J* = 4.4, 0.9 Hz, H-6), 6.86 (1H, d, *J* = 4.1 Hz, H-2); ^13^C-NMR (MeOH-d_4_, *150 MHz*, ppm) δ 48.6 (CH, C-6), 52.8 (CH_3_, COO*Me*), 67.6 (CH, C5), 69.3 (CH, C3), 76.2 (CH, C4), 131.8 (Cq, C1), 141.6 (CH, C-2), 167.3 (Cq, *C*OOMe); HREIMS m/z calcd for C_8_H_10_O_5_^79^Br [M − H]^+^ 264.9712, found 264.9713.

#### 3.1.5. Synthesis of Methyl (−)-3,4-*O*-Cyclohexylidene-3,4,5-Trihydroxy-6-iodo-1-CycloHexene Carboxylate (10_I_) 

To a solution of (−)-**8** (14.4 mg, 0.056 mmol) in CH_2_Cl_2_ (1.0 mL) was added AlI_3_ (7.9 mg, 0.019 mmol) at 0 °C with stirring. After stirring for 2 h at 0 °C, the reaction was quenched by adding sat. NaHCO_3_ (aq) (5 mL). The resulting mixture was then extracted with CH_2_Cl_2_ (3 × 10 mL). The combined organic layers were dried over MgSO_4_, filtered, and evaporated to afford the crude product, which was purified by silica gel column chromatography (eluent = Hexane:EtOAc = 5:1) to afford (−)-**10**_I_ (15.5 mg, 60%) as a white powder. mp 97–99 °C; [α]_D_^20^ –261.4 (*c* 0.09, CHCl_3_); ^1^H-NMR (CDCl_3_, *600 MHz*, ppm) δ 1.43–1.74 (10H, m), 2.70 (1H, d, *J* = 2.3 Hz, 5-OH), 3.82 (3H, s, -OMe), 4.36 (1H, ddd, *J* = 4.1, 3.2, 2.3 Hz, H-5), 4.74 (1H, dd, *J* = 7.9, 3.2 Hz, H-3), 4.86 (1H, dd, *J* = 7.9, 4.1 Hz, H-4), 5.21 (1H, d, *J* = 3.2 Hz, H-6), 7.10 (1H, d, *J* = 3.2 Hz, H-2); ^13^C-NMR (CDCl_3_, *150 MHz*, ppm) δ 16.8 (CH_2_), 23.5 (CH_2_), 24.0 (CH_2_). 25.1 (CH_2_), 33.3 (CH_2_), 36.0 (CH, C-6), 52.5 (CH_3_, COO*Me*), 68.0 (CH, C-5), 69.1 (CH, C-3), 72.6 (CH, C-4), 110.4 (Cq), 132.5 (Cq, C-1), 135.8 (CH, C-2), 164.7 (Cq, *C*OOMe); HREIMS *m/z* calcd for C_14_H_10_O_5_I [M]^+^ 394.0277, found 394.0279.

(+)-**10_I_** (17.6 mg, 72%) was synthesized from (+)-**8** (16.5 mg, 0.062 mmol) in a similar manner as (−)-**10_I_**. White powder; mp 97–100 °C; [α]_D_^20^ +262.0 (*c* 0.13, CHCl_3_); ^1^H-NMR (CDCl_3_, *400 MHz*, ppm) δ 1.43–1.74 (10H, m), 2.71 (1H, d, *J* = 1.6 Hz, 5-OH), 3.82 (3H, s, -OMe), 4.34–4.38 (1H, m, H-5), 4.74 (1H, dd, *J* = 7.8, 3.1 Hz, H-3), 4.86 (1H, dd, *J* = 7.8, 3.5 Hz, H-4), 5.21 (1H, d, *J* = 3.3 Hz, H-6), 7.10 (1H, d, *J* = 3.2 Hz, H-2); ^13^C-NMR (CDCl_3_, *100 MHz*, ppm) δ 16.8, 23.5, 23.9. 25.0, 33.2, 35.9, 52.5, 68.0, 69.1, 72.5, 110.4, 132.4, 135.8, 164.6; HREIMS *m/z* calcd for C_14_H_10_O_5_I [M]^+^ 394.0277, found 394.0279.

#### 3.1.6. Synthesis of (−)-6-Iodoopericosine A (1_I_)

To a solution of (−)-**10**_I_ (4.3 mg, 0.011 mmol) in MeOH (1.0 mL) was added TFA (4.5 mL) at 0 °C with stirring. The reaction mixture was stirred for another 3 h at rt, then concentrated under reduced pressure give a crude residue, which was purified by column chromatography with neutral silica gel (eluent = 5% MeOH in CH_2_Cl_2_) to afford (−)**-1_F_** as a colorless oil (2.6 mg, 76%). [α]_D_^20^ −8.4 (*c* 0.13, MeOH); ^1^H-NMR (acetone-d_6_, *600 MHz*, ppm) δ 3.79 (3H, s, -OMe), 4.21 (1H, dd, *J* = 4.7, 2.1 Hz, H-4), 4.33 (1H, br dd, *J* = 3.5, 2.1 Hz, H-5), 4.43 (1H, br t, *J* = 4.7, H-3), 5.16 (1H, dd, *J* = 3.5, 0.6 Hz, H-6), 6.88 (1H, d, *J* = 4.4 Hz, H-2); ^13^C-NMR (acetone-d_6_, *150 MHz*, ppm) δ 26.8 (CH, C-6), 52.6 (CH_3_, COO*C*H_3_), 66.6 (CH, C5), 67.7 (CH, C-4), 76.9 (CH, C-5), 132.8 (Cq, C1), 139.5 (CH, C-2), 166.4 (Cq, *C*OOMe); HREIMS *m/z* calcd for C_8_H_11_O_5_I [M]^+^ 313.9652, found 313.9649.

(+)-**1_I_** (1.9 mg, 77%)was synthesized from (+)-**10_I_** (3.1 mg, 0.0079 mmol) in a similar manner as (−)-**1_I_**; Colorless oil; [α]_D_^20^ +11.7 (*c* 0.10, MeOH); ^1^H-NMR (acetone-d_6_, *600 MHz*, ppm) δ 3.79 (3H, s, -OMe), 4.21 (1H, dd, *J* = 4.7, 2.1 Hz, H-4), 4.33 (1H, br dd, *J* = 3.5, 2.1 Hz, H-5), 4.43 (1H, br t, *J* = 4.7, H-3), 5.16 (1H, dd, *J* = 3.5, 0.6 Hz, H-6), 6.88 (1H, d, *J* = 4.4 Hz, H-2); ^13^C-NMR (acetone-d_6_, *150 MHz*, ppm) δ 26.8 (CH, C-6), 52.6 (CH_3_, COO*C*H_3_), 66.7 (CH, C-3), 67.7 (CH, C-4), 76.9 (CH, C-5), 132.8 (Cq, C1), 139.5 (CH, C-2), 166.4 (Cq, *C*OOMe); HREIMS *m/z* calcd for C_8_H_11_O_5_I [M]^+^ 313.9652, found 313.9655.

### 3.2. Synthesis of Pericoxide

#### Synthesis of Methyl (3R,4R,5R,6S)-5-Bromo-3,4,6-Trihydroxycyclohex-1-ene-1-carBoxylate (11) from 9

A solution of **9** (264 mg, 0.76 mmol) in dry MeOH (4.0 mL) in a microwave (MW) vial was added to Dowex (760 mg). The vial was then sealed and heated under MW irradiation at 100 °C for 30 min. After cooling, the reaction mixture was filtered, the filtrate concentrated under reduced pressure to give the crude product, which was purified via column chromatography (eluent: CH_2_Cl_2_:MeOH = 95: 5) to afford **11** (164 mg, 80%). [α]_D_^20^ −71.1 (c 1.285, MeOH); IR (liquid film) ν_max_ 3417 (OH), 1714 (C=O), 1650 (C=C) cm^−1^; ^1^H-NMR (CD_3_OD, *600 MHz*, ppm) δ 3.78 (3H, s, OMe), 3.84 (1H, dd, *J* = 8.2, 4.1 Hz, H-4), 4.29 (1H, dd, *J* = 8.2, 5.3 Hz, H-5), 4.47 (1H, t, *J* = 4.1 Hz, H-3), 4.64 (1H, d, *J* = 5.3 Hz, H-6), 6.77 (1H, dd, *J* = 4.1, 0.6 Hz, H-2); ^13^C-NMR (CD_3_OD, *150 MHz*, ppm) δ 52.5 (COO*Me*-C8), 55.1 (CH, C5), 67.0 (CH, C3), 71.4 (CH, C4), 71.5 (CH, C6), 138.6 (CH, C2), 167.8 (*C*OOMe-C7); HREIMS *m*/*z* calcd for C_8_H_12_O_5_^79^Br_1_ [M + H]^+^ 266.9868 found 266.9865, C_8_H_12_O_5_^81^Br_1_ [M + H]^+^ 268.9848 found 268.9850.

### 3.3. Intramolecular Epoxidation of Bromotriol 11

To a solution of 1,1,1,3,3,3-hexamethyldisilazane (HMDS) (0.22 mL, 1.07 mmol) in THF (5 mL) was added *n*-BuLi (0.66 mL, 1.06 mmol) at −78 °C to prepare a solution of lithium hexamethyldisilazide (LHMDS). After stirring for 30 min, the LHMDS solution was added dropwise to a solution of **11** (92.7 mg, 0.35 mmol) in THF (5 mL) at −78 °C. After stirring the reaction mixture at −78 °C for 1 h, the resulting mixture was warmed to rt and stirred for another 1 h. The reaction mixture was then treated with sat. NH_4_Cl aq. (20 mL) and extracted with EtOAc (3 × 20 mL). The combined organic layers were dried over MgSO_4_, filtered, and evaporated to give the crude product, which was purified by column chromatography (CH_2_Cl_2_: MeOH = 95: 5) to afford methyl (1*S*,2*S*,5*R*,6*R*)-2,5-dihydroxy-7-oxabicyclo[4.1.0]hept-3-ene-3-carboxylate (**12**) (27.4 mg, 42%). [α]_D_^20^ −27.2 (c 0.29, MeOH); IR (liquid film) ν_max_ 3391 (OH), 1713 (C=O), 1655 (C=C) cm^−1^; ^1^H-NMR (CD_3_OD, *600 MHz*, ppm) δ 3.05 (1H, ddd, *J* = 4.0, 2.6, 0.9, H-6), 3.54 (1H, dd, *J* = 4.1 2.9 Hz, H-1), 3.76 (3H, s, COO*Me*), 4.49 (1H, dd, *J* = 4.4, 2.3 Hz, H-5), 4.58 (1H, s, OH), 4.68 (1H, dd, *J* = 2.6, 1.2 Hz, H-2), 6.43 (1H, dd, *J* = 4.1, 2.3 Hz, H-4); ^13^C-NMR (CD_3_OD, *150 MHz*, ppm) δ 52.4 (CH_3_, COO*Me*), 55.4 (CH, C-6), 56.0 (CH, C-1), 64.1 (CH, C-2), 65.9 (CH, C-5), 131.5 (Cq, C-3), 137.8 (CH, C-4),168.0 (Cq, *C*OOMe); HREIMS *m*/*z* calcd for C_8_H_11_O_5_ [M + H]^+^ 187.0606 found 187.0604.

### 3.4. Synthesis of (−)-Pericoxide (7)

A solution of 1.6 M *n*-BuLi in hexane (0.17 mL, 0.27 mmol) was added to a solution of HMDS (0.089 mL, 0.43 mmol) in THF (2 mL) at −78 °C with stirring to prepare an LHMDS solution. After 40 min, the prepared LHMDS solution was added to a solution of **10**_Br_ (22.1 mg, 0.085 mmol) in THF (2 mL) at −78 °C. After stirring for 1.5 h at −78 °C, the reaction mixture was quenched with sat. NH_4_Cl aq. (20 mL) and extracted with EtOAc (7 × 20 mL). The combined organic layers were dried over MgSO_4_, filtered, and concentrated under reduced pressure to give the crude product, which was purified via column chromatography (eluent; CH_2_Cl_2_:MeOH = 97:3) using neutral silica gel (DIOL MB100–75/200, Fiji Silysia Co.) to afford (+)-**1** (11.9 mg, 77%); Colorless oil; [α]_D_^20^ −68.3 (c 0.60, MeOH); IR (film) ν_max_ 3414 (OH), 1720 (C=O), 1651 (C=C) cm^−1^; ^1^H-NMR (CD_3_OD, *600 MHz*, ppm) δ 3.63 (1H, ddd, *J* = 4.4, 2.1, 1.5 Hz, H-5), 3.81 (3H, s, COO*Me*), 3.97 (1H, d, *J* = 4.1 Hz, H-6), 3.98 (1H, d, *J* = 4.4 Hz, H-4), 4.21 (1H, ddd, *J* = 6.2, 5.0, 2.1 Hz, H-3), 4.58 (1H, s, OH), 7.12 (1H, dd, *J* = 6.5, 2.4 Hz, H-2); ^13^C-NMR (CD_3_OD, *150 MHz*, ppm) δ 50.1 (CH-C6), 52.8 (COO*Me*-C8), 58.7 (CH-C5), 66.6 (CH-C3), 68.6 (CH-C4), 132.0 (CH, C-1), 143.3 (CH, C-2), 167.3 (Cq, *C*OOMe); HREIMS *m/z* calcd for C_8_H_10_O_5_ [M]^+^ 186.0528, found 186.0523.

(+)-**7** (3.7 mg, 87%) was synthesized from (+)-**10**_Br_ (6.2 mg, 0.023 mmol) in the same manner. Colorless oil; [α]_D_^20^ +65.8 (c 0.60, MeOH); ^1^H-NMR (CD_3_OD, *600 MHz*, ppm) δ 3.63 (1H, ddd, *J* = 4.4, 2.1, 1.5 Hz, H-5), 3.81 (3H, s, COO*Me*), 3.96–3.98 (2H, m, H-4, H-6), 4.21 (1H, ddd, *J* = 6.2, 5.0, 2.1 Hz, H-3), 4.57 (1H, s, OH), 7.12 (1H, dd, *J* = 6.5, 2.4 Hz, H-2); ^13^C-NMR (CD_3_OD, *150 MHz*, ppm) δ 50.1 (CH-C6), 52.8 (COO*Me*-C8), 58.7 (CH-C5), 66.6 (CH-C3), 68.6 (CH-C4), 132.0 (CH, C-1), 143.3 (CH, C-2), 167.2 (Cq, *C*OOMe), *some degradation was detected during measuring the ^13^C-NMR spectrum; HREIMS *m/z* calcd for C_8_H_10_O_5_ (M)^+^ 186.0528, found 186.0525; Literature data of (+)-**7 [34]**: [α]_D_ +74 (*c* = 0.13, MeOH); ^1^H-NMR (CD_3_OD, *400 MHz*, ppm) δ 3.64 (1H, m, H-5), 3.81 (3H, s, COO*Me*), 3.96 (1H, m, H-6), 3.98 (1H, m, H-4), 4.21 (1H, m, H-3), 4.58 (1H, s, OH), 7.10 (1H, dd, *J* = 6.4, 2. 3 Hz, H-2); ^13^C-NMR (CD_3_OD, *100 MHz*, ppm) δ 50.1 (CH, C-6), 52.8 (CH_3_, COO*Me*), 58.6 (CH, C-5), 66.4 (CH, C-3), 68.5 (CH, C-4), 131.2 (CH, C-1), 143.2 (CH-2), 167.2 (Cq, C-7); HRESIMS *m*/*z* calcd for C_8_H_10_O_5_Na [M + Na]^+^ 209.0420, found 209.0416.

### 3.5. Biological Assay

Antitumor and glucosidase inhibitory assays were performed using the same procedures as those described in our previous paper [35].

## 4. Conclusions

The synthesis of both enantiomers of pericosine A analogs bearing F, Br, and I atoms was achieved for the first time and their antitumor activity against P388, L1210, and HL-60 cell lines was evaluated. Although all of the synthesized compounds were moderately active against the three types of tumor cell lines studied, significant differences between their enantiomers and differences between the halogens, except for fluorine, were not observed. The fluorinated derivatives showed weaker activities than the other analogs and pericosine A.

Form glycosidase inhibitory assay, five synthesized molecules of **1** except for (+)-**1**_F_ were elucidated to exhibit α-glucosidase inhibitory activity at mM level of IC_50._ As well, (−)-**1**_Br_ and (+)-**1**_I_ showed inhibitory activities against β-galactosidase and α-galactosidase, respectively, at mM level of IC_50_.

In addition, both enantiomers of pericoxide were synthesized using 6-bromopericosine A as a suitable synthetic precursor.

## Data Availability

Not applicable.

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
