# Peer review of "Synthesis of 6-Halo-Substituted Pericosine A and an Evaluation of Their Antitumor and Antiglycosidase Activities"

_marinedrugs, 2022, doi:10.3390/md20070438_

Round 1

Reviewer 1 Report

In this paper the authors reported the synthesis of pericosine A analogs, focusing on the introduction of halogen atoms on C6 cyclohexene ring system. The novel compounds have been tested both as anticancer agents and as glycosidase inhibitors. Notwithstanding this work appears incremental, in my opinion it is eligible to be published on Marine Drugs as the authors have described in detail the synthetic steps for obtaining the new compounds and have directed efforts towards their optimization. Chemistry reported herein and in previous works can be useful to scientists working on the functionalization of cyclohexene skeletons.

Major concerns

1. The manuscript is overall well written; however, for a better readability they are asked to add the known reactions in the synthetic schemes.

2. In the SI enhance the intensity of signals in NMR spectra and provide figures with a better resolution.

3. Please provide 1H NMR spectra of pure compounds (for example the compound (-)-10F is contaminated by several impurities).

4. For the fluorinated compounds add inset of 19F NMR spectra in the 1H NMR spectra

5. Conclusions seem a bit hasty.

6. In the biological section, the authors should explain the effect of the halogen atoms on the observed cytotoxicity.

Minor concerns

1. In the Scheme 1 R is H.

2. Check the consistence of numbering of the cyclohexene ring system (Figure 1(a) and Scheme 2(a)).

3. I think is glycosidase inhibitory activities and not anti-glycosidase inhibitory activities.

4. Line 75: required; line 81: equivalents.

Author Response

1. The manuscript is overall well written; however, for a better readability they are asked to add the known reactions in the synthetic schemes.

Response: Thank you for suggestion. We made least modification with a small addition of reaction condition. Some multi-steps transformation are abbreviated because detailed reactions had been already reported in provided references.  

2. In the SI enhance the intensity of signals in NMR spectra and provide figures with a better resolution.

Response: We think resolution is good enough. Too much resolution led sized to the file larger. Attached original JPEG file are 200 dpi size.

3. Please provide 1H NMR spectra of pure compounds (for example the compound (-)-10F is contaminated by several impurities).

Response: We changed 1H- and 13C-NMRspectra for (-)-10F

4. For the fluorinated compounds add inset of 19F NMR spectra in the 1H NMR spectra.

 Response: Thank you for suggestion. But, we consumed all samples for biological assay unfortunately. We need couple of weeks to synthesize them. So we are sorry that we cannot provide them in limited period. We think 19F-NMR might be informative as the reviewer suggested but many journals do not require them.

5. Conclusions seem a bit hasty.

  Response: Thank you for suggestion. We revised conclusion.

6. In the biological section, the authors should explain the effect of the halogen atoms on the observed cytotoxicity.

Response: Thank you for suggestion. We revised text.

7. In the Scheme 1 R is H.

Response: We revised as suggestion.

8. Check the consistence of numbering of the cyclohexene ring system (Figure 1(a) and Scheme 2(a)).

Response: Thank you for suggestion. But numbering of cychexene ring in Figure 1 is based on pericosine core in references #13 and 14. And numbering in Figure 2 we noted is based on IUPAC rule.

9. I think is glycosidase inhibitory activities and not anti-glycosidase inhibitory activities.

Response: Thank you for suggestion: We revised anti-glycosidase inhibitory activities to glycosidase inhibitory activities.

10. Line 75: required; line 81: equivalents.

 Response: Thank you for suggestion. We revised as suggested.

Reviewer 2 Report

This work is part of an ongoing research of the Authors focused on the synthesis and biological properties of the pericosine analogues (see refs. 31, 32, 35). In the present manuscript they describe the synthesis of both anomers of 6-halogenated pericosine A that showed moderate antitumor activity and rather modest glycosidase inhibition activity (IC50 in the millimolar range).

The work is interesting and deserves publication, however, the presentation of the results must be improved. In particular, it is extremely difficult to follow and understand the long series of reactions reported at pages 3-5. Why did they show all the syntheses of 6-halogenated pericosine A in a single scheme (Scheme 1)? This scheme should be divided in a few smaller ones and the discussion should be put just before the relevant scheme. Moreover, I did not understand whether they used in the present work the "commercially available HBr and HI aqueous solutions" as stated at page 2, line72.

In the Experimental section it is not clear at all which ionisation technique was employed for the High-Resolution mass spectra of the isolated compounds. In fact, sometimes it is written just "HRMS" and the calculated value was for (M)+ (does it mean that electron impact was used?), sometimes it is written "HREIMS" and the calculated value was for (M)+ or (M+H)+, sometimes it is written "HRMS" and the calculated value was for (M+H)+ (does it mean that electrospray ionisation was used?). Please, clarify this issue.

Author Response

1. It is extremely difficult to follow and understand the long series of reactions reported at pages 3-5. Why did they show all the syntheses of 6-halogenated pericosine A in a single scheme (Scheme 1)? This scheme should be divided in a few smaller ones and the discussion should be put just before the relevant scheme.

Response: Thank you for suggestion. We made least modification on Scheme 1. New reactions to each target molecules is only 2 steps from common intermediate compound 8. For easy finding, common intermediate compound 8 was enclosed in square in Scheme 1. Some multi-steps transformation are abbreviated because detailed reactions had been already reported in provided references. So, we combined in one Scheme. We are sorry to the reviewer for inconvenience.

2. I did not understand whether they used in the present work the "commercially available HBr and HI aqueous solutions" as stated at page 2, line72.

Response: We did not use them in the present work.

3. In the Experimental section it is not clear at all which ionisation technique was employed for the High-Resolution mass spectra of the isolated compounds. In fact, sometimes it is written just "HRMS" and the calculated value was for (M)+ (does it mean that electron impact was used?), sometimes it is written "HREIMS" and the calculated value was for (M)+ or (M+H)+, sometimes it is written "HRMS" and the calculated value was for (M+H)+ (does it mean that electrospray ionisation was used?). Please, clarify this issue.

Response: Thank you for suggestion. We revised along reviewer’s suggestion.

Round 2

Reviewer 1 Report

The authors have partially addressed the questions raised.

I asked to explain the effect of the halogen atoms on the cytotoxicity, and the authors replied saying: "Because pericosine C (3), which exists as an enantiomeric mixture in nature and has the same relative configuration to 1s, was reported to  be inactive against P-388 cell line [14], present results implied importance of the presence of halogen atom at C-6 in pericosine core structure for antitumor activity". 

I asked to add informative 19F NMR spectra and the authors replied:  Thank you for suggestion. But, we consumed all samples for biological assay unfortunately. We need couple of weeks to synthesise them. So we are sorry that we cannot provide them in limited period. We think 19F-NMR might be informative as the reviewer suggested but many journals do not require them.

I asked to check the consistency of the numbering, as the epoxide 12 may be considered a pericosine analogue

Reviewer 2 Report

Some corrections have been made although the Scheme 1 has been only slightly changed and the relevant discussion has not been modified.